# UNSUPERVISED TASK CLUSTERING FOR MULTI-TASK REINFORCEMENT LEARNING

## ABSTRACT

Meta-learning, transfer learning and multi-task learning have recently laid a path towards more generally applicable reinforcement learning agents that are not limited to a single task. However, most existing approaches implicitly assume a uniform similarity between tasks. We argue that this assumption is limiting in settings where the relationship between tasks is unknown a-priori. In this work, we propose a general approach to automatically cluster together similar tasks during training. Our method, inspired by the expectation-maximization algorithm, succeeds at finding clusters of related tasks and uses these to improve sample complexity. We achieve this by designing an agent with multiple policies. In the expectation step, we evaluate the performance of the policies on all tasks and assign each task to the best performing policy. In the maximization step, each policy trains by sampling tasks from its assigned set. This method is intuitive, simple to implement and orthogonal to other multi-task learning algorithms. We show the generality of our approach by evaluating on simple discrete and continuous control tasks, as well as complex bipedal walker tasks and Atari games. Results show improvements in sample complexity as well as a more general applicability when compared to other approaches.

## 1 INTRODUCTION

Imagine we are given an arbitrary set of tasks. We know that dissimilarities and/or contradicting objectives can exist. However, in most settings we can only guess these relationships and how they might affect joint training. Many recent works rely on such human guesses and (implicitly or explicitly) limit the generality of their approaches. This can lead to impressive results, either by explicitly modeling the relationships between tasks as in transfer learning (Zhu et al., 2020), or by meta learning implicit relations (Hospedales et al., 2020). However, in some cases an incorrect similarity assumption can hurt learning performance (Lazaric, 2012). Our aim with this paper is to provide an easy, straightforward approach to avoid human assumptions on task similarities.

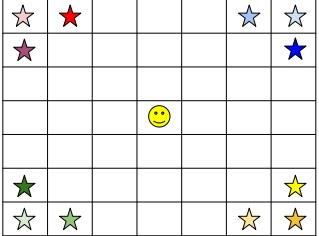

Figure 1: An agent (smiley) should reach one of 12 goals (stars) in a grid world. Learning to reach a goal in the top right corner helps him to learn about the other goals in that corner. However, learning to reach the green stars (bottom left corner) at the same time gives conflicting objectives, hindering training. Task clustering resolves the issue.

An obvious solution is to train a separate policy for each task. However, this leads to a large amount of experience being required to learn the desired behaviors. Therefore, it is desirable to have a single agent and allow the sharing of knowledge between tasks. This is generally known as multi-task learning, a field which has received a large amount of interest in both the supervised learning and reinforcement learning (RL) community (Zhang & Yang, 2017). If tasks are sufficiently similar, a policy that is trained on one task provides a good starting point for another task, and experience from each task will help training in the other tasks. This is known as *positive transfer* (Lazaric, 2012). However, if the tasks are sufficiently dissimilar, *negative transfer* occurs and reusing a pre-trained policy is disadvantageous. It can even lead to a worse performance than simply starting with a random initialization. Here using experience from the other tasks might slow training or even prevent con-

vergence to a good policy. Most previous approaches to multi-task learning do not account for problems caused by negative transfer directly and either accept its occurrence or limit their experiments to sufficiently similar tasks. We present a hybrid approach that is helpful in a setting where the task set contains clusters of related tasks, amongst which transfer is helpful. To illustrate the intuition we provide a conceptualized example in Figure 1. The figure shows a grid world with 12 tasks that can be naturally clustered in 4 clusters. Note however that our approach goes beyond this conceptual ideal and can be beneficial even if the clustering is not perceivable by humans a-priori.

Our approach is inspired by the expectation-maximization framework and uses a set of completely separate policies within our agent. We iteratively evaluate the set of policies on all tasks, assign tasks to policies based on their respective performance and train policies on their assigned tasks. This leads to policies naturally specializing to clusters of related tasks, yielding an interpretable decomposition of the full task set. Moreover, we show that our approach can improve the learning speed and final reward in multi-task RL settings. To summarize our contributions:

- We propose a general approach inspired by Expectation-Maximization (EM) that can find clusters of related tasks in an unsupervised manner during training.

- We provide an evaluation on a diverse set of multi-task RL problems that shows the improved sample complexity and reduction in negative transfer in our approach.

- We show the importance of meaningful clustering and the sensitivity to the assumed number of clusters in an ablation study

## 2 RELATED WORK

Expectation-Maximization (EM) has previously been used in RL to directly learn a policy. By reformulating RL as an inference problem with a latent variable, it is possible to use EM to find the maximum likelihood solution, corresponding to the optimal policy. We direct the reader to Deisenroth et al. (2013) for a survey on the topic. Our approach is different: We use an EM-inspired approach to cluster tasks in a multi-task setting and rely on recent RL algorithms to learn the tasks.

In supervised learning, the idea of subdividing tasks into related clusters was proposed by Thrun & O'Sullivan (1996). They use a distance metric based on generalization accuracy to cluster tasks. Another popular idea related to our approach that emerged from supervised learning is the use of a mixture of experts (Jacobs et al., 1991). Here, multiple sub-networks are trained together with an input dependent gating network. Jordan & Jacobs (1993) also proposed an EM algorithm to learn the mixture of experts. While those approaches have been extended to the control setting (Jacobs & Jordan, 1990; 1993; Meila & Jordan, 1995; Cacciatore & Nowlan, 1993; Tang & Hauser, 2019), they rely on an explicit supervision signal. It is not clear how such an approach would work in an RL setting. A variety of other methods have been proposed in the supervised learning literature, for brevity we direct the reader to the survey by Zhang & Yang (2017), which provides a good overview of the topic. Our work differs in that we focus on RL, where no labeled data set exists.

In RL, task clustering has in the past received attention in works on transfer learning. Carroll & Seppi (2005) proposed to cluster tasks based on a distance function. They propose distances based on $Q$-values, reward functions, optimal policies or transfer performance. They propose to use the clustering to guide transfer. Similarly, Mahmud et al. (2013) propose a method for clustering Markov Decision Processes (MDPs) for source task selection. They design a cost function for their chosen transfer method and derive an algorithm to find a clustering that minimizes this cost function. Our approach differs from both in that we do not assume knowledge of the underlying MDPs and corresponding optimal policies. Furthermore, the general nature of our approach allows it to scale to complex tasks, where comparing properties of the full underlying MDPs is not feasible. An earlier approach by Wilson et al. (2007) developed a hierarchical Bayesian approach for multi-task RL. Their approach uses a Dirichlet process to cluster the distributions from which they sample full MDPs in the hope that the sampled MDP aligns with the task at hand. They then solve the sampled MDP and use the resulting policy to gather data from the environment and refine the posterior distributions for a next iteration. While their method is therefore limited to simple MDPs, our approach can be combined with function approximation and therefore has the potential to scale to MDPs with large or infinite state spaces which cannot be solved in closed form. Lazaric & Ghavamzadeh (2010) use a hierarchical Bayesian approach to infer the parameters of a linear value function and utilize

EM to infer a policy. However, as this approach requires the value function to be a linear function of some state representation, this approach is also difficult to scale to larger problems which we look at. Li et al. (2009) note that believe states in partially observable MDPs can be grouped according to the decision they require. Their model infers the parameters of the corresponding decision state MDP. Their approach scales quadratically with the number of decision states and at least linearly with the number of collected transitions, making it as well difficult to apply it to complex tasks.

More recent related research on multi-task RL can be split into two categories: works that focus on very similar tasks with small differences in dynamics and reward, and works that focus on very dissimilar tasks. In the first setting, approaches have been proposed that condition the policy on task characteristics identified during execution. Lee et al. (2020) use model-based RL and a learned embedding over the local dynamics as additional input to their model. Yang et al. (2020) train two policies, one that behaves in a way that allows the easy identification of the environment dynamics and another policy that uses an embedding over the transitions generated by the first as additional input. Zintgraf et al. (2020) train an embedding over the dynamics that accounts for uncertainty over the current task during execution and condition their policy on it. Our approach is more general than these methods as our assumption on task similarity is weaker. In the second group of papers, the set of tasks is more diverse. Most approaches here are searching for a way to reuse representations from one task in the others. Riemer et al. (2018) present an approach to learn hierarchical options, and use it to train an agent on 21 Atari tasks. They use the common NatureDQN network (Mnih et al., 2015) with separate final layers for option selection policies, as well as separate output layers for each task to account for the different action spaces. Eramo et al. (2020) show how a shared representation can speed up training. They then use a network strucuture with separate heads for each task, but shared hidden layers. Our multi-head baseline is based on these works. Bräm et al. (2019) propose a method that addresses negative transfer between multiple tasks by learning an attention mechanism over multiple sub-networks, similar to a mixture of expert. However, as all tasks yield experience for one overarching network, their approach still suffers from interference between tasks. We limit this interference by completely separating policies. Wang et al. (2020) address the problem of open-ended learning in RL by iteratively generating new environments. Similar to us, they use policy rankings as a measure of difference between tasks. However, they use this ranking as a measure of novelty to find new tasks, addressing a very different problem. Hessel et al. (2019) present PopArt for multi-task deep RL. They address the issue that different tasks may have significantly different reward scales. Sharma et al. (2018) look into active learning for multi-task RL on Atari tasks. They show that uniformly sampling new tasks is suboptimal and propose different sampling techniques. Yu et al. (2020) propose Gradient Surgery, a way of projecting the gradients from different tasks to avoid interference. These last three approaches are orthogonal to our work and can be combined with EM-clustering. We see this as an interesting direction for future work.

Quality-Diversity (QD) algorithms (Pugh et al., 2016; Cully & Demiris, 2018) in genetic algorithms research aim to find a diverse set of good solutions for a given problem. One proposed benefit of QD is that it can overcome local optima by using the solutions as "stepping stones" towards a global optimum. Relatedly in RL, Eysenbach et al. (2018) and Achiam et al. (2018) also first identify diverse skills and then use the learned skills to solve a given task. While we do not explicitly encourage diversity in our approach, our approach is related in that our training leads to multiple good performing, distinct policies trained on distinct tasks. This can lead to a policy trained on one task becoming the best on a task that it was not trained on, similar to the "stepping stones" in QD. However, in our work this is more a side-effect than the proposed functionality.

## 3 BACKGROUND AND NOTATION

In RL (Sutton & Barto, 2017) tasks are specified by a Markov Decision Process (MDP), defined as tuple $(\boldsymbol{S}, \boldsymbol{A}, P, R, \gamma)$, with state space $\boldsymbol{S}$, action space $\boldsymbol{A}$, transition function $P(\cdot|s, a)$, reward function $R(s, a)$ and decay factor $\gamma$. As we are interested in reusing policies for different tasks, we require a shared state-space $\boldsymbol{S}$ and action-space $\boldsymbol{A}$ across tasks. Note however that this requirement can be omitted by allowing for task specific layers. Following prior work, we do allow for a task specific final layer in our Atari experiments to account for the different action spaces. In all other experiments however, tasks only differ in their transition function and reward function. We therefore describe a task as $\tau = (P_\tau, R_\tau)$ and refer to the set of given tasks as $\mathcal{T}$. For each task $\tau \in \mathcal{T}$ we aim to maximize the discounted return $G_\tau = \sum_{t=0}^{t=L} \gamma^t r_t^\tau$, where $r_t^\tau \sim R_\tau(s_t, a_t)$ is the reward at

time step $t$ and $L$ is the episode length. Given a set of policies $\{\pi_1, ..., \pi_n\}$, we denote the return obtained by policy $\pi_i$ on task $\tau$ as $G_\tau(\pi_i)$.

## 4 CLUSTERED MULTI-TASK LEARNING

As the growing body of literature on meta-, transfer- and multi-task learning suggests, we can expect a gain through positive transfer if we train a single policy $\pi_i$ on a set of related tasks $\mathcal{T}_k \subset \mathcal{T}$. On the flip side, the policy $\pi_i$ might perform poorly on tasks $\tau \notin \mathcal{T}_k$. Moreover, training policy $\pi_i$ on a task $\tau \notin \mathcal{T}_k$ might even lead to a decrease in performance on the task set $\mathcal{T}_k$ through negative transfer. We incorporate these insights into our algorithm by modeling the task set $\mathcal{T}$ as a union of $K$ disjoint task clusters $\mathcal{T}_1, \ldots, \mathcal{T}_K$, i.e., $\mathcal{T} = \bigcup_{k=1}^{K} \mathcal{T}_k$ with $\mathcal{T}_i \cap \mathcal{T}_j = \emptyset$ for $i \neq j$. Tasks within a cluster allow for positive transfer while we do not assume any relationship between tasks of different clusters. Tasks in different clusters may therefore even have conflicting objectives. Note that the assignment of tasks to clusters is not given to us and therefore needs to be inferred by the algorithm. Note also that this formulation only relies on minimalistic assumptions. That is, we do not assume a shared transition function or a shared reward structure. Neither do we assume the underlying MDP to be finite and/or solvable in closed form. Our approach is therefore applicable to a much broader range of settings than many sophisticated models with stronger assumptions. As generality is one of our main objectives, we see the minimalistic nature of the model as a strength rather than a weakness.

---

**Algorithm 1:** EM-Task-Clustering

Initialize $n$ policies $\{\pi_1, ..., \pi_n\}$
**while** not converged **do**
    ▷ E-Step
    $\mathcal{T}_i \leftarrow \emptyset$ for $i \in \{1, \ldots, n\}$
    **for** $\tau \in \mathcal{T}$ **do**
        $k \leftarrow \arg\max_i G_\tau(\pi_i)$
        $\mathcal{T}_k \leftarrow \mathcal{T}_k \cup \tau$
    $\mathcal{T}_i \leftarrow \mathcal{T}$ where $\mathcal{T}_i = \emptyset$
    ▷ M-Step
    **for** $\pi_i \in \{\pi_1, ..., \pi_n\}$ **do**
        $t \leftarrow 0$
        **while** $t < T_M$ **do**
            $\tau \sim \mathcal{T}_i$
            Train $\pi_i$ on $\tau$ for an episode of $L$ steps
            $t \leftarrow t + L$

---

Given this problem formulation we note that it reflects a clustering problem, in which we have to assign each task $\tau \in \mathcal{T}$ to one of the clusters $\mathcal{T}_k$, $k \in \{1, \ldots, K\}$. At the same time, we want to train a set of policies $\{\pi_1, ..., \pi_n\}$ to solve the given tasks. Put differently, we wish to infer a hidden latent variable (cluster assignment of the tasks) while optimizing our model parameters (set of policies). An Expectation-Maximization (EM) (Dempster et al., 1977) inspired algorithm allows us to do exactly that. On a high level, in the expectation step (E-step) we assign each of the tasks $\tau \in \mathcal{T}$ to a policy $\pi_i$ representing cluster $\mathcal{T}_i$. We then train the policies in the maximization step (M-step) on the tasks they got assigned, specializing the policies to their clusters. These steps are alternatingly repeated — one benefiting from the improvement of the other in the preceding step — until convergence. Given this general framework we are left with filling in the details. Specifically, how to assign tasks to which policies (E-step) and how to allocate training time from policies to assigned tasks (M-step).

For the assignment in the E-step we want the resulting clusters to represent clusters with positive transfer. Given that policy $\pi_i$ is trained on a set of tasks $\mathcal{T}_i$ in a preceding M-step, we can base our assignment of tasks to $\pi_i$ on the performance of $\pi_i$: Tasks on which $\pi_i$ performs well likely benefited from the preceding training and therefore should be assigned to the cluster of $\pi_i$. Specifically, we can evaluate each policy $\pi_i \in \{\pi_1, \ldots, \pi_n\}$ on all tasks $\tau \in \mathcal{T}$ to get an estimate of $G_\tau(\pi_i)$ and base the assignment on this performance evaluation. To get to an implementable algorithm we state two additional desiderata for our assignment: (1) We do not want to constrain cluster sizes in any way as clusters can be of unknown, non-uniform sizes. (2) We do not want to constrain the diversity of the tasks. This implies that the assignment has to be independent of the reward scales of the tasks, which in turn limits us to assignments based on the relative performances of the policies $\pi_1, ..., \pi_n$. We found a greedy assignment — assigning each task to the policy that performs best — to work well. That is, a task $\tau_k$ is assigned to the policy $\pi = \arg\max_{\pi_i} G_{\tau_k}(\pi_i)$. A soft assignment based on the full ranking of policies might be worth exploring in future work. Given the greedy assignment, our method can also be seen as related to k-means (MacQueen, 1967), a special case of EM.

In the M-step, we take advantage of the fact that clusters reflect positive transfer, i.e., training on some of the assigned tasks should improve performance on the whole cluster. We can therefore

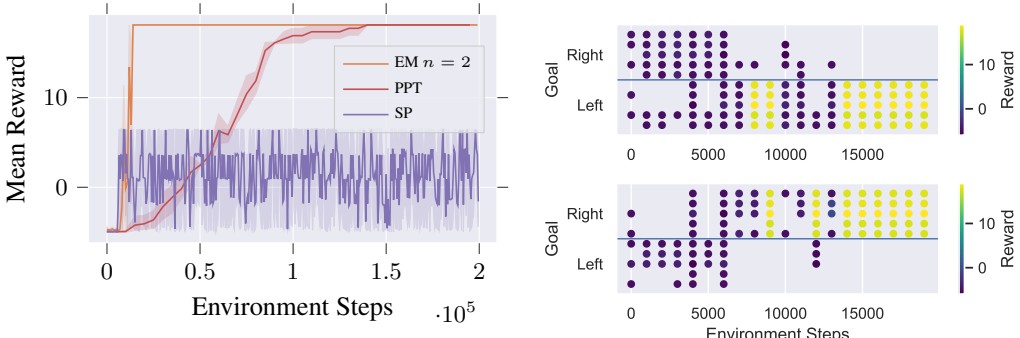

Figure 2: **Left:** Mean reward and 95% confidence interval (shaded area) from 10 trials when training on the chain environment. **Right:** Task assignment (dots) and task specific reward (color) over the course of training the two policies in our approach. Each plot shows one of the policies/estimated clusters. The assignments converge to the natural clustering reflected by the goal location.

randomly sample a task from the assigned tasks and train on it for one episode before sampling the next task. Overall we train each policy for a fixed number of updates $T_M$ in each M-step with $T_M$ independent of the cluster size. This independence allows us to save environment interactions as larger clusters benefit from positive transfer and do not need training time proportional to the number of assigned tasks.

Note that the greedy assignment (and more generally any assignment fulfilling desiderata 1 above) comes with a caveat: Some policies might not be assigned any tasks. In this case we sample the tasks to train these policies from all tasks $\tau \in \mathcal{T}$, which can be seen as a random exploration of possible task clusters. This also ensures that, early on in training, every policy gets a similar amount of initial experience. For reference, we provide a simplified pseudo code of our approach in Algorithm 1. Note that we start by performing an E-Step, i.e., the first assignment to clusters is based on the performance of the randomly initialized policies. Note also that our approach is independent of the RL algorithm used to train the policies in the M-step and can therefore be combined with any state-of-the-art RL algorithm.

## 5 EXPERIMENTS

As a proof of concept we start the evaluation of our approach on two discrete tasks. The first environment consists of a chain of discrete states in which the agent can either move to the left or to the right. The goal of the agent is placed either on the left end or the right end of the chain. This gives rise to two task clusters, where tasks within a cluster differ in the frequency with which the agent is rewarded on its way to the goal. The second environment reflects the 2-dimensional grid-world presented in Figure 1. Actions correspond to the cardinal directions in which the agent can move and the 12 tasks in the task set $\mathcal{T}$ are defined by their respective goal. We refer an interested reader to Appendix A.1 for a detailed description of the environments.[1]

We train policies with tabular Q-learning (Watkins, 1989) and compare our approach to two baselines: In the first we train a single policy on all tasks. We refer to this as SP (Single Policy). In the other we train a separate policy per task and evaluate each policy on the task it was trained on. This is referred to as PPT (Policy per Task). Our approach is referred to as EM (Expectation-Maximization).

The learning curves as well as the task assignment over the course of training are shown in Figure 2 and Figure 3. Looking at the assignments, we see that in both environments our approach converges to the natural clustering, leading to a higher reward after finding these assignments. Both our EM-approach and PPT converge to an optimal reward in the chain environment, and a close to optimal reward in the corner-grid-world. However, PPT requires a significantly higher amount of environment steps to reach this performance, as it does not share information between tasks and therefore

---

[1]The implementation of all our experiments is also available in the supplementary material

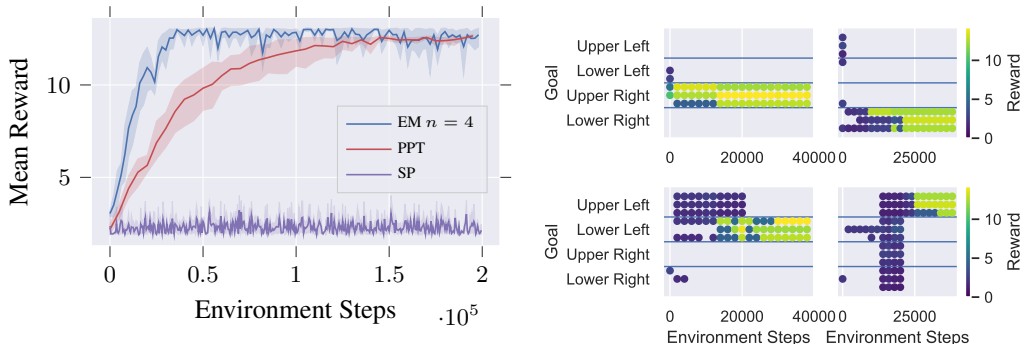

Figure 3: **Left:** Mean reward and 95% confidence interval (shaded area) from 10 trials when training on the grid-world environment depicted in Figure 1. **Right:** Task assignment (dots) and task specific reward (color) over the course of training for the $n = 4$ policies (estimated clusters) in our approach. The assignment naturally clusters the tasks of each corner together.

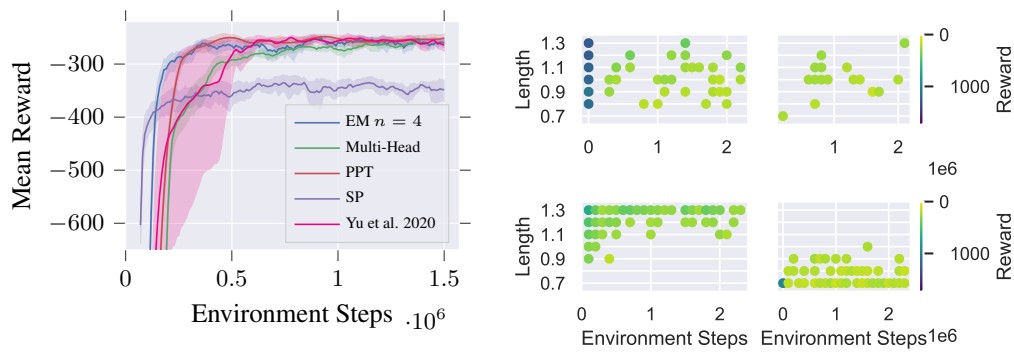

Figure 4: **Left:** Mean reward and 95% confidence interval (shaded area) from 10 trials when training on the pendulum environment. The curves are smoothed by a rolling average to dampen the noise of the random starting positions. For (Yu et al., 2020) we used 12 trials out of which 3 failed to converge and were excluded. **Right:** Task assignment (dots) and task specific reward (color) from a sample run. Two policies focus on long and short, while the others focus on medium lengths.

has to do exploration for each task separately. SP fails to achieve a high reward due to the different tasks providing contradicting objectives.

## 5.1 PENDULUM

Next we consider a simple continuous control environment where tasks differ in their dynamics. We use the pendulum gym task (Brockman et al., 2016), in which a torque has to be applied to a pendulum to keep it upright. Here the environment is the same in all tasks, except for the length of the pendulum, which is varied in the range $\{0.7, 0.8, ..., 1.3\}$, giving a total of 7 tasks. Note that there is no clear cluster boundary here.

We use Twin Delayed Deep Deterministic Policy Gradient (TD3) (Fujimoto et al., 2018) with hyperparameters optimized as discussed in Appendix A.2. We use $n = 4$ policies in all experiments (except for the ablation studies) and did not tune this hyperparameter. This was done to give a fair comparison to baseline approaches which do not have this extra degree of freedom. For application purposes the number of clusters can be treated as a hyperparameter and included in the hyperparameter optimization. We compare against a SP, PPT, gradient surgery Yu et al. (2020) and a multi-head network structure similar to the approach used by Eramo et al. (2020). Each policy in our approach uses a separate replay buffer. The multi-head network has a separate replay-buffer and a separate input and output layer per task. We adjust the network size of the multi-head baseline and SP to avoid an advantage of our method due to a higher parameter count, see Appendix A.2 for details.

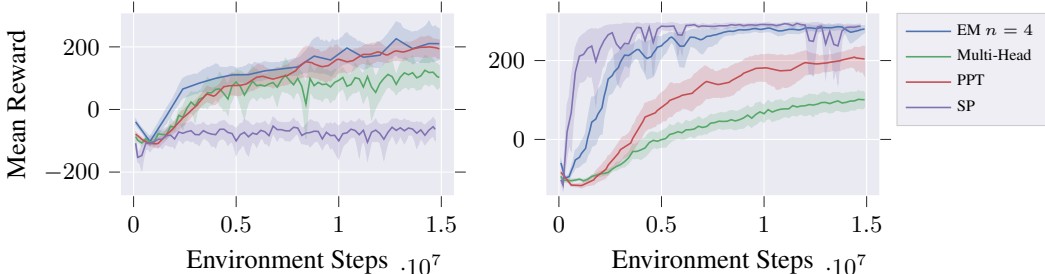

Figure 5: Evaluation of the BipedalWalker experiments. The shaded areas show the 95% confidence interval on the mean task reward. **Left:** Track and field task set; 6 tasks with varying objectives. Results reflect 20 trials of each approach. **Right:** Task set with varying leg lengths and obstacles; 9 tasks with the same reward function. Results reflect 10 trials of each approach.

The results are shown in Figure 4. In the figure we excluded 3 from the 12 runs of the gradient surgery baseline as those did not converge to a performance better than random. Still, the results show a worse sample complexity and higher variance compared to our approach. We also again observe that our approach clusters similar tasks together, leading to a better performance than with a SP agent, and a faster convergence than with PPT. Also the multi-head approach needs more experience to converge than our approach in this setup, even more than the PPT approach. We believe this is due to the inherent interference of learning signals in the shared layers. The cluster assignment in our approach is also intuitive, with two clusters focusing on the extremes (cf. Figure 4).

## 5.2 BIPEDAL WALKER

As a more complex continuous control environment we focus on *BipedalWalker* from the OpenAI Gym (Brockman et al., 2016), which has previously been used in multi-task and generalization literature (Portelas et al., 2019; Wang et al., 2019; 2020). It consists of a bipedal robot in a two-dimensional world, where the default task is to move to the right with a high velocity. The action space consists of continuous torques for the hip and knee joints of the legs and the state space consists of joint angles and velocities, as well as hull angle and velocity and 10 lidar distance measurements.

To test our approach, we designed 6 tasks inspired by track and field sports: Jumping up at the starting position, jumping forward as far as possible, a short, medium and long run and a hurdle run. As a second experiment, we create a set of 9 tasks by varying the leg length of the robot as well as the number of obstacles in its way. This task set is inspired by task sets in previous work (Portelas et al., 2019). Note that we keep the objective — move forward as fast as possible — constant here. We again use TD3 and tune the hyperparameters of the multi-head baseline and our approach (with $n = 4$ fixed) with grid-search. Experiment details and hyperparameters are given in Appendix A.3.

The results in Figure 5 (left) on the track and field tasks show a significant advantage in using our approach over multi-head TD3 or SP and a slightly better initial performance than PPT, with similar final performance. SP fails to learn a successful policy altogether due to the conflicting reward functions. In contrast, the results in Figure 5 (right) from the second task set show that SP can learn a policy that is close to optimal on all tasks here. The multi-head and PPT approaches suffer in this setup as each head/policy only gets the experience from its task and therefore needs more time to converge. Our approach can take advantage of the similarity of the tasks. We note that the experiments presented here reflect two distinct cases: One in which it is advantageous to separate learning, reflected by PPT outperforming SP, and one where it is better to share experience between tasks, reflected by SP outperforming PPT. Our approach demonstrates general applicability as it is the only one performing competitively in both. We provide an insight into the assignment of tasks to policies in Appendix B.1.

## 5.3 ATARI

To test the performance of our approach on a more diverse set of tasks, we evaluate on a subset of the Arcade Learning Environment (ALE) tasks (Machado et al., 2018). Our choice of tasks is similar to

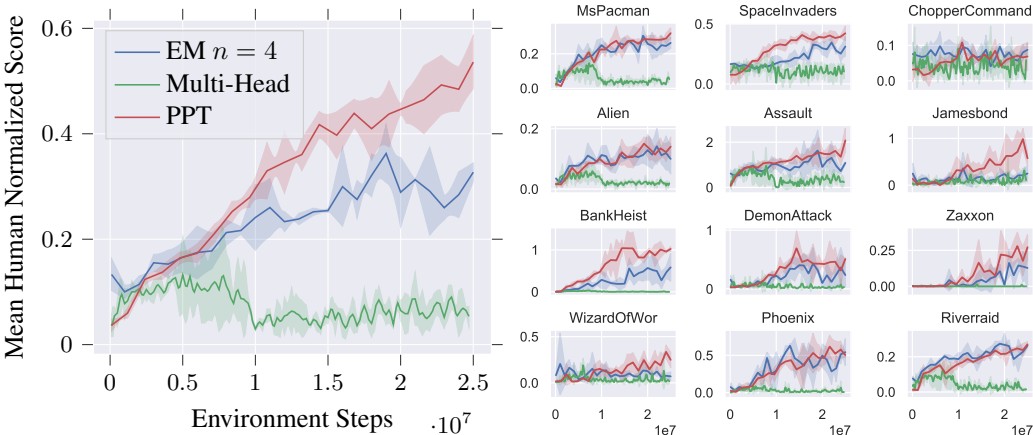

Figure 6: The results of our experiments on a subset of the Atari Learning Environment games. The reward is averaged across 3 trials and the shaded region shows the standard deviation of the mean.

those used by Riemer et al. (2018), but we exclude tasks containing significant partial-observability. This is done to reduce the computational burden as those tasks usually require significantly more training data. We built our approach on top of the Implicit Quantile Network (IQN) implementation in the Dopamine framework (Dabney et al., 2018; Castro et al., 2018). We chose IQN due to its sample efficiency and the availability of an easily modifiable implementation. As the different ALE games have different discrete action spaces, we use a separate final layer and a separate replay buffer for each game in all approaches. We use the hyperparameters recommended by Castro et al. (2018), except for a smaller replay buffer size to reduce memory requirements. As in the Bipedal Walker experiments we fix the number of policies in our approach without tuning to $n = 4$. We choose the size of the network such that each approach has the same number of total tunable parameters. We provide the details in Appendix A.4.

The results are given in Figure 6. The good performance of PPT shows that the diversity of this task set is best addressed with a policy per task. Note that we did not expect our approach to outperform PPT here as our approach can only outperform PPT if there are similar tasks that can be clustered together. The diverse set of Atari games seems to violate this assumption. We note also that the multi-head approach is unable to learn any useful policy here due to negative transfer between tasks. This is in line with experiments in other research (Hessel et al., 2019). Our approach manages to overcome most of this negative interference, even with just 4 clusters. Task assignments in our approach are given in Appendix B.2.

## 5.4 ABLATIONS

To gain additional insight into our approach, we perform two ablation studies on the discrete corner-grid-world environment and the pendulum environment.

First, we investigate the performance of our approach for different numbers of policies $n$. The results in Figure 7 show that using too few policies can lead to a worse performance, as the clusters cannot distinguish the contradicting objectives. On the other hand, using more policies than necessary increases the number of environment interactions required to achieve a good performance in the pendulum task, but does not significantly affect the final performance.

As a second ablation, we are interested in the effectiveness of the clustering. It might be possible that simply having fewer tasks per policy is giving our approach an advantage compared to SP or multi-head TD3. We therefore provide an ablation in which task-policy assignments are determined randomly at the start and kept constant during the training. Results from this experiment can be seen in Figure 8, with additional results in Appendix C. The results show that using random clusters performs significantly worse than using the learned clusters. This highlights the importance of clustering tasks meaningfully.

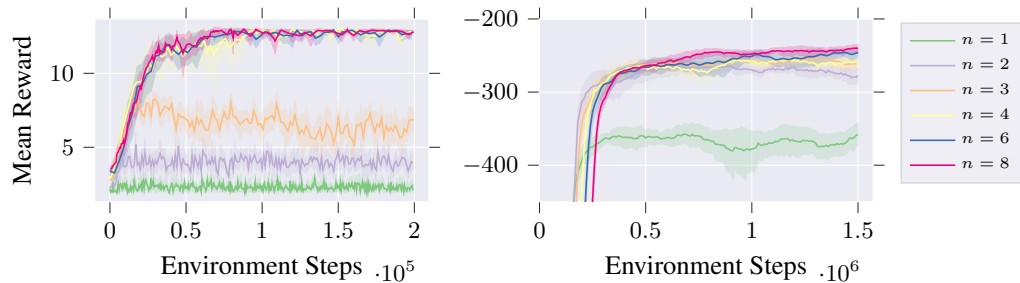

Figure 7: Ablations for different number of policies $n$. Shaded areas show the 95% confidence interval of the mean reward from 10 trials each. **Left:** Corner-grid-world tasks. **Right:** Pendulum tasks, learning curves smoothed.

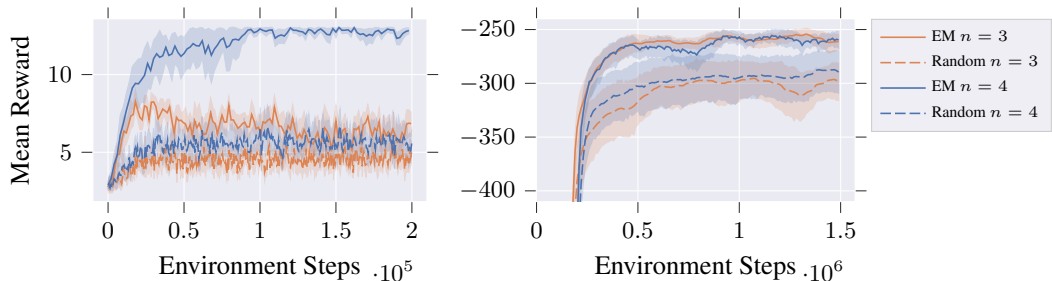

Figure 8: Comparison of our approach against randomly assigning tasks to policies at the start of training. Shaded areas show the 95% confidence interval of the mean reward. **Left:** Corner-grid-world tasks, 10 trials each. **Right:** Pendulum tasks, 10 trials each, learning curves smoothed.

## 6 CONCLUSION

We present an approach for multi-task learning in reinforcement learning (RL) that automatically clusters tasks into related subsets. Our approach uses a set of policies and alternatingly evaluates the policies on all tasks, assigning each task to the best policy and then trains policies on their assigned tasks. Since our approach can be combined with any underlying RL method, we evaluate it on a varied set of environments. We show its performance on sets of simple discrete tasks, simple continuous control tasks, two complex continuous control task sets and a set of Arcade Learning Environment tasks. We show that our approach is able to identify clusters of related tasks and use this structure to achieve a competitive or superior performance. We further provide an ablation over the number of policies in our approach, showing that too many policies can lead to slower convergence while too few policies can hurt performance. In another ablation we also highlight the need to cluster tasks meaningfully.

Our approach offers many possibilities for future extensions. One interesting direction would be hierarchical clustering. This could prove helpful for complicated tasks like the Atari games. It would also be interesting to see how our approach can be applied to multi-task learning in a supervised setting. Further, different assignment strategies with soft assignments could be investigated. Overall, we see our work as a good stepping stone for future work on structured multi-task learning.

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

APPENDIX

## A  EXPERIMENT DETAILS

In addition to the details provided here, the implementation of all experiments can be found in the supplementary material.

### A.1  GRID WORLD EXPERIMENTS

In the first discrete task set we use a one-dimensional state-chain with 51 states, in which the agent starts in the middle and receives a reward for moving toward either the left or right end. As a reward we use $r = \frac{1}{|x_{\mathrm{ag}} - x_{\mathrm{goal}}|}$ where $x_{\mathrm{ag}}$ is the position of the agent and $x_{\mathrm{goal}}$ is the goal position (either the left or right end of the chain). We give a reward of $r = 20$ if the goal position is reached. Depending on the task, the reward is given every 2, 4, 8 or 16 steps, or only at the goal position, and otherwise replaced by $r = 0$.

For our corner grid-world task set we use a 2D-grid-world with edge length 7 and three goal positions per corner (as depicted in Figure 1). The agent always starts in the center and receives a reward based on the distance to the target $r = \frac{1}{||x_{\mathrm{ag}} - x_{\mathrm{goal}}||_2}$, with $|| \cdot ||_2$ being the Euclidean norm. A reward of $r = 10$ is given when the agent reaches the goal position.

In both tasks we use tabular Q-Learning with $\epsilon$-greedy exploration. We start with $\epsilon_0 = 0.2$ and decay the value as $\epsilon_t = \epsilon_0^{\gamma_\epsilon^t}$ with $\gamma_\epsilon = 1 - 1 \times 10^{-6}$. We use a learning rate of $\alpha = 0.2$ to update the value estimates, as from the perspective of a single agent the environment can be regarded as stochastic. Further, we use a discount factor of $\gamma = 0.9$ and $T_M = 500$ training steps per policy in each M-step and evaluate each policy on each task for three episodes during the E-step, using the greedy policy without exploration.

### A.2  PENDULUM

In our pendulum tasks we use a modified version of the Pendulum environment provided in OpenAI gym (Brockman et al., 2016). This environment consists of a single pendulum and the goal is to balance it in an upright position. The observation consists of the current angle $\theta$, measured from the upright position, and current angular velocity represented as $(\sin\theta, \cos\theta, \dot{\theta})$. The reward for each time step is $r_t = -(\theta^2 + 0.1\dot{\theta}^2 + 0.001a^2)$, with $a$ being the torque used as action. Every episode starts with a random position and velocity. To provide a set of tasks we vary the length of the pendulum in $\{0.7, 0.8, ..., 1.3\}$.

#### A.2.1  HYPERPARAMETERS

Hyperparameters for our EM-TD3 and multi-head TD3 were tuned on the pendulum task set by grid search over learning rate $\alpha = \{1 \times 10^{-2}, 3 \times 10^{-3}, 1 \times 10^{-3}\}$, batch-size $b = \{64, 128\}$ and update-rate $u = \{1, 3, 5\}$, specifying the number of collected time-steps after which the value-function is updated. We increased the network size for multi-head TD3, so that it overall had more parameters than EM-TD3. This is done to eliminate a potential advantage of our approach stemming from a higher representational capacity. The tuned hyperparameters are given in Table 1. To represent the value functions and policies we use fully connected multi-layer perceptrons (MLPs) with two hidden layers with 64 units each. As activations we use ReLU on all intermediate layers, and tanh activations on the output. The values are then scaled to the torque limits per dimension. In EM, SP and PPT we use a separate network for each policy. For our multi-head baseline we share the hidden layers between tasks, but use separate input and output layers per task. Additionally, we increase the size of the first hidden layer to 96 in the multi-head approach, such that it has a similar total number of parameters as our EM approach. For SP and PPT we reuse the hyper-parameters from our EM approach. During the M-step, we train the agent for $5 \times 10^4$ steps per policy and during the E-step we evaluate each agent on each task by running 20 episodes without added exploration noise.

Table 1: Hyperparamters for pendulum experiments.

| Hyperparameter | EM-TD3 | Multi-head TD3 |
|---|---|---|
| learning-rate $\alpha$ | $3 \times 10^{-3}$ | $3 \times 10^{-3}$ |
| batch-size $b$ | 128 | 128 |
| update-rate $u$ | 1 | 1 |
| policy-update-frequency | 3 | 3 |
| $n$ - EM | 4 | - |
| network size | $4 \cdot (64, 64, 1)$ | $(9 \cdot 96, 64, 9 \cdot 1)$ |
| exploration noise $\sigma$ | 0.05 | 0.05 |
| exploration noise clipping | $[-0.5, 0.5]$ | $[-0.5, 0.5]$ |
| target policy smoothing noise $\sigma$ | 0.1 | 0.1 |
| buffer-size | $2e6$ per policy | $2e6$ per task |
| decay $\gamma$ | 0.99 | 0.99 |
| $T_M$ | $5 \times 10^4$ | - |

## A.3 BIPEDALWALKER

For the BipedalWalker tasks we look at two different sets of tasks. The first set of tasks consists of different reward functions with mostly similar environments, inspired by track and field events. The tasks are jumping up, jumping a long distance, runs for different distances and a run with obstacles. In all tasks a reward of $-\epsilon||a||_1$ is given to minimize the used energy. The position of the hull of the bipedal walker is denoted as $(x, y)$. In the jump up task a reward of $y - |x|$ is given upon landing, and $\epsilon = 3.5 \times 10^{-4}$. For the long jump task a reward of $x - x_0$ is given upon landing, with $x_0$ being the hull position during the last ground contact, $\epsilon = 3.5 \times 10^{-4}$. The three runs consist of a sprint over a length of 67 units, with $\epsilon = 3.5 \times 10^{-4}$, a run over 100 units, with $\epsilon = 3.5 \times 10^{-4}$, and a long run over 200 units with $\epsilon = 6.5 \times 10^{-4}$. The hurdles task is identical to the long run, but every 4 units there is an obstacle with a height of 1. Additionally, a reward of $0.1\dot{x}$ — a reward proportional to the velocity of the agent in the x-direction — is given during the run and hurdle tasks, to reward movement to the right.

The second set of tasks consists of varying obstacles and robot parameters. We vary the length of the legs in $\{25, 35, 45\}$ and either use no obstacles, or obstacles with a spacing of 2 or 4 units apart and height of 1. This results in a total of 9 tasks. Here we use the standard reward for the BipedalWalker task $r = 4.3\dot{x} - 5|\theta| - ||a||_1$ with $\theta$ being the angle of the walker head. Additionally, in all experiments $r = -100$ is given if the robot falls over or moves to far to the left.

### A.3.1 HYPERPARAMETERS

Hyperparameters for our EM-TD3 and multi-head TD3 approaches were tuned on the track and field task set by grid search over $\alpha = \{1 \times 10^{-3}, 3 \times 10^{-4}, 1 \times 10^{-4}\}$, batch-size $b = \{100, 1000\}$ and update-rate $u = \{1, 3, 5\}$, $u$ specifying the number of collected time-steps after which the value-function is updated. We reuse the optimal parameters found here on the task set with varying leg lengths and obstacles. For the SP and PPT baselines we reused the parameters from EM-TD3. We increased the network size for multi-head TD3, so that it overall had more parameters than EM-TD3. All hyperparameters are given in Table 2. During the M-step, we train the EM agent with $2 \times 10^5$ steps per policy and during the E-step we evaluate each agent on each task by running 20 episodes without added exploration noise.

## A.4 ATARI

To test our approach on a more complex task, we evaluate it on a subset of the Atari games. The set of chosen games consists of Alien, Assault, BankHeist, ChopperCommand, DemonAttack, James-Bond, MsPacman, Phoenix, RiverRaid, SpaceInvaders, WizardOfWor and Zaxxon. As stated above, this task set is similar to the set of games used in Riemer et al. (2018), but without tasks requiring a large amount of exploration to save computation time.

Table 2: Hyperparameters for BipedalWalker experiments.

| Hyperparameter | EM-TD3 | Multi-head TD3 |
|---|---|---|
| learning-rate | $1 \times 10^{-3}$ | $1 \times 10^{-3}$ |
| batch-size | 1000 | 1000 |
| update-rate | 3 | 5 |
| policy-update-frequency | 3 | 3 |
| $n$ - EM | 4 | - |
| network size | $4 \cdot (400, 300, 1)$ | $(6 \cdot 400, 400, 6 \cdot 1)$ |
| exploration noise $\sigma$ | 0.1 | 0.1 |
| exploration noise clipping | $[-0.5, 0.5]$ | $[-0.5, 0.5]$ |
| target policy smoothing noise $\sigma$ | 0.2 | 0.2 |
| buffer-size | $5e6$ per policy | $5e6$ per task |
| decay $\gamma$ | 0.99 | 0.99 |
| $T_M$ | $2 \times 10^5$ | - |

Our implementation is based on the IQN implementation in the Dopamine framework (Dabney et al., 2018; Castro et al., 2018). As hyperparameters we use the default values recommended by Dopamine for Atari games, except the changes listed below: Due to the different action spaces, we use a separate replay buffer for each game, as well as a separate output layer, both for our EM, multi-head and PPT approaches. We reduce the size of the replay buffer to $3 \times 10^5$ compared to $1 \times 10^6$ in the original paper, to reduce the memory demand. We use the normal NatureDQN network, but scale the size of the layers to ensure that each approach has a similar number of parameters. For our EM approach, we use $T_M = 2.5 \times 10^5$ trainings steps per M-step, and evaluate all policies on all tasks for 27000 steps in the E-step, using the greedy policy without random exploration. In both EM and the multi-head approach, we record how many transitions were performed in each M-Step and sample the task with the least transitions as next training task. This is done to ensure a similar amount of transitions and training steps per game, as episode lengths vary. This approach was proposed in Riemer et al. (2018).

## B  ADDITIONAL RESULTS

### B.1  BIPEDAL WALKER

In Figure 9 the assignments for 4 randomly chosen trials on the track and field task set are shown. We can see that in all trials the runs over different distances are grouped together with the long jump task. This is likely due to these tasks aligning well, as they both favor movements to the right. It is possible to learn the hurdles task with the same policy as the runs, due to the available LIDAR inputs. The hurdle task therefore sometimes switches between policies, but usually is learned by a separate policy. The jump up task is very different from the other tasks, as it is the only one not to involve movement to the right, and is therefore assigned to a separate policy.

In Figure 10 the assignments for 4 randomly chosen trials on the leg-length and obstacle task set are shown. As illustrated by the good performance of the SP approach shown in Figure 5, it is possible to learn a nearly optimal behavior with a single policy here. This makes learning a meaningful clustering significantly harder and sometimes leads to a single policy becoming close to optimal on all tasks, as in Trial 2. In most other trials the task set is separated into two or three different clusters based on the different leg lengths.

### B.2  ATARI

In Figure 11 the assignments of all three trials of our approach on the Atari task set are shown. While we see a consistency in assignments, we cannot identify a clearly repeated clustering across trial. We assume this is due to the high diversity of tasks preventing the identification of clearly distinguishable clusters. This lack of clearly distinguishable clusters might also be the reason for failing to reach the performance of PPT. Yet, the specialization of policies in our approach helps to avoid negative transfer as seen in Figure 6.

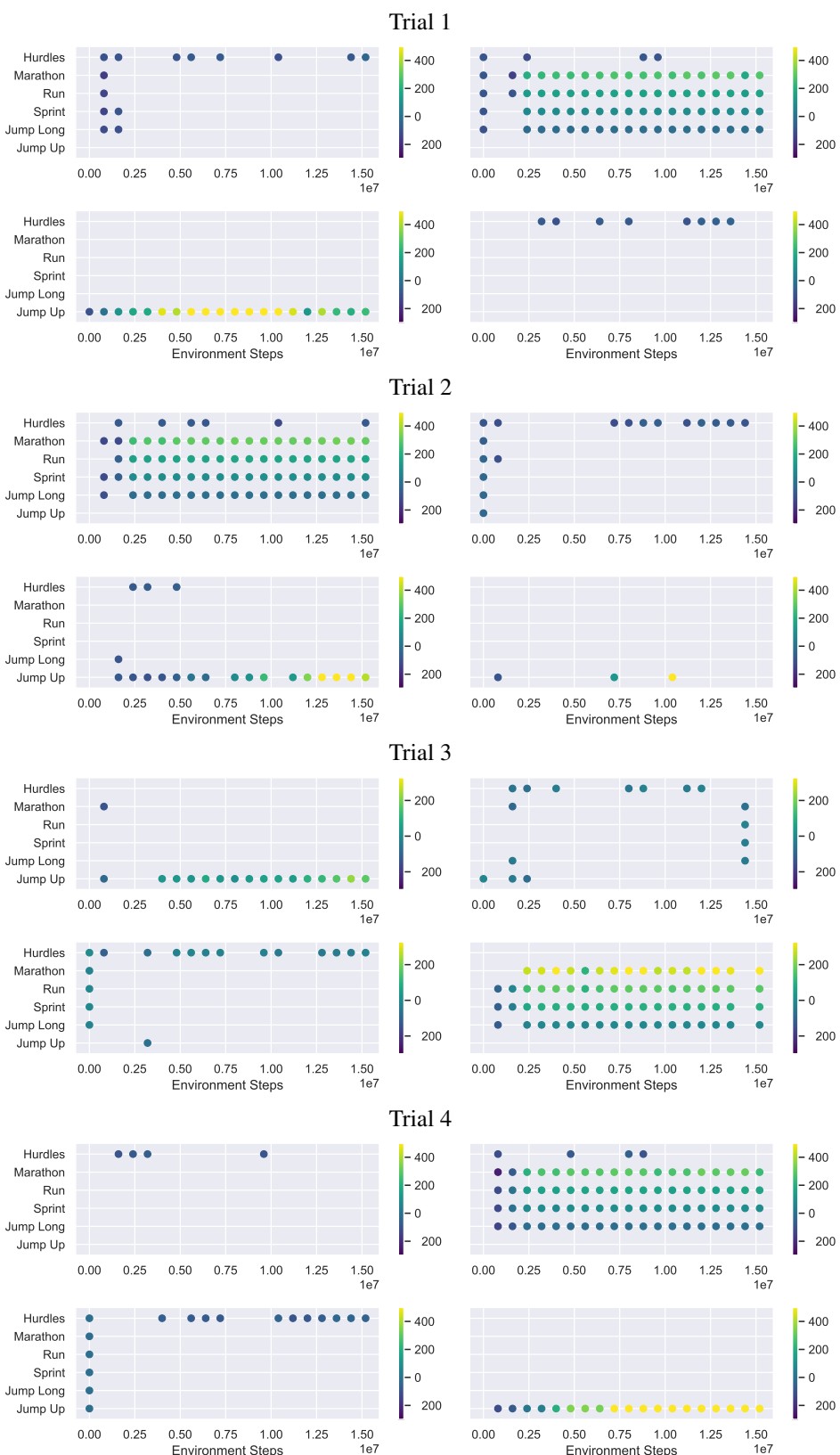

Figure 9: Shown are the assignments from 4 randomly picked trials on the track and field Bipedal-Walker task set.

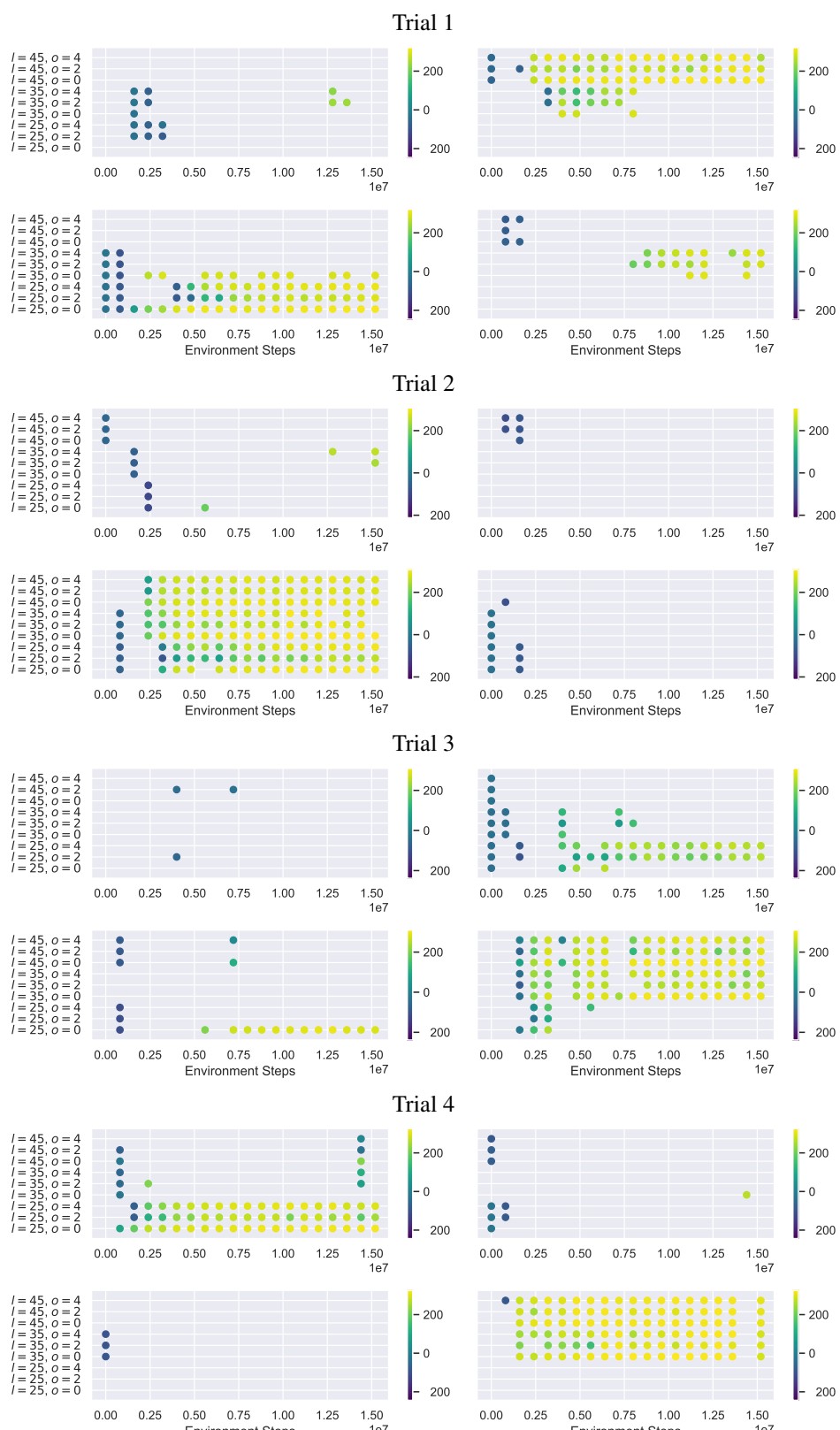

Figure 10: Shown are the assignments from 4 randomly picked trials on the first BipedalWalker task set. $l$ refers to the lenghts of the legs, $o$ refers to the frequency of obstacles.

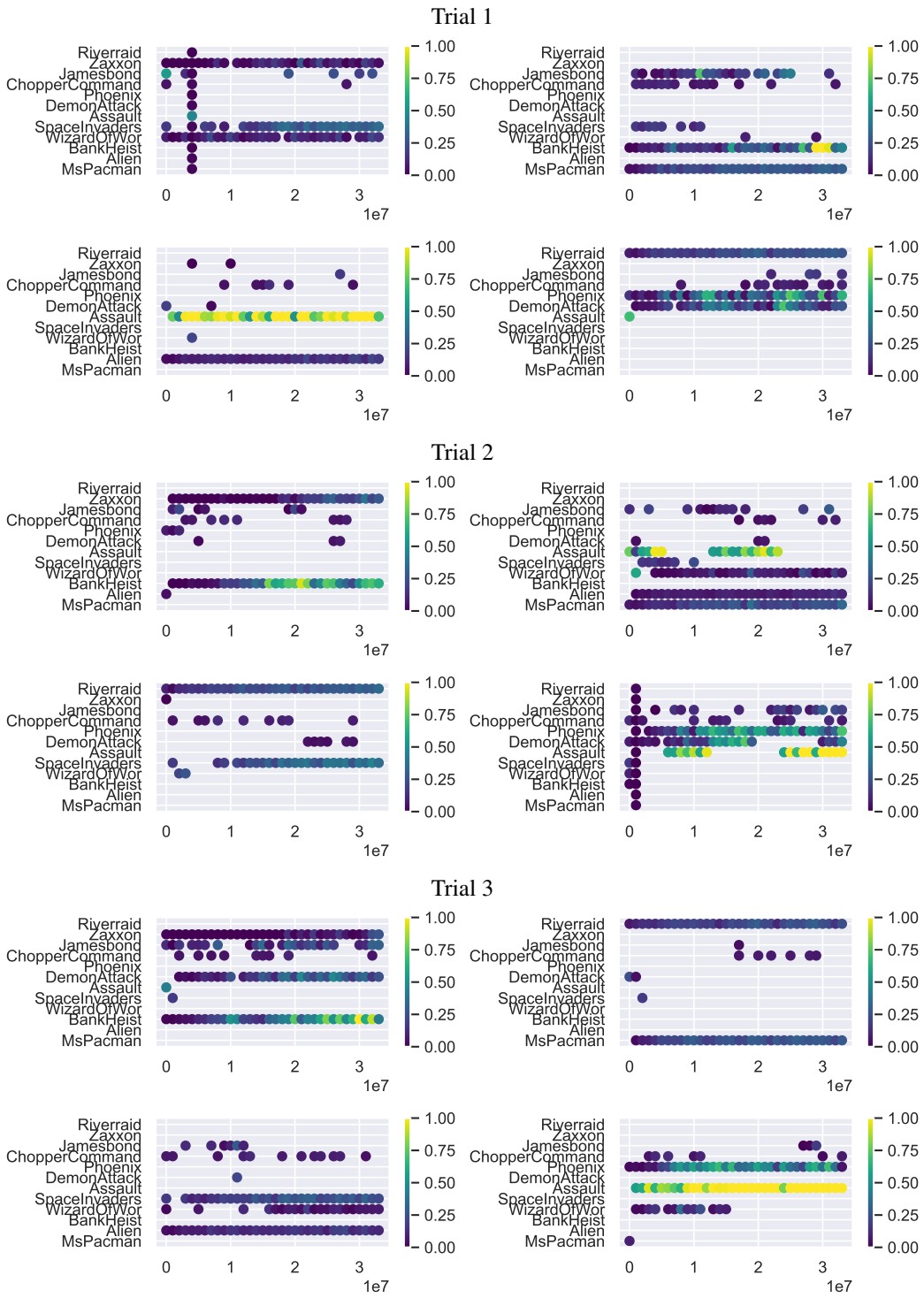

Figure 11: Shown are the assignments of all three trial that were run on the set of Atari games. The color represents the human-normalized score per game.

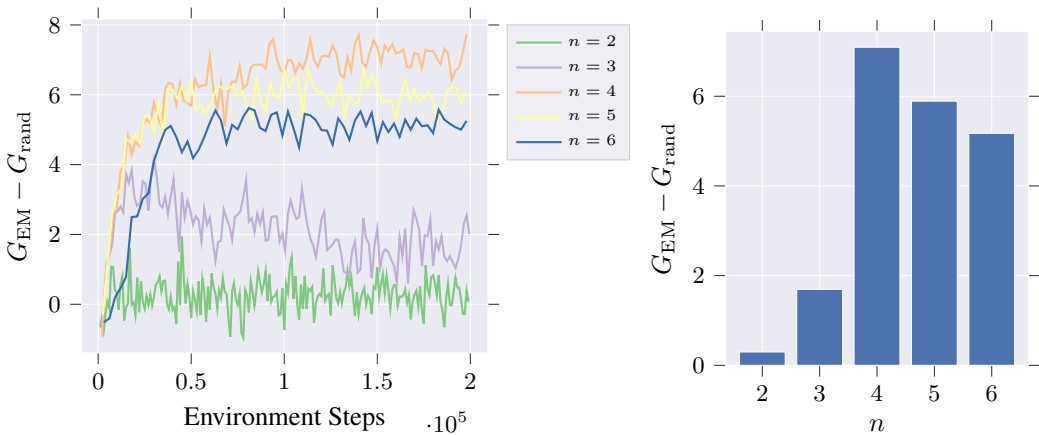

Figure 12: Show is the difference between using random assignments or our EM approach for different numbers of policies. On the left the development during training is shown, on the right the average performance gap over the last 10% of the training is visualised.

## C    PERFORMANCE GAP TO RANDOM CLUSTERS

In Figure 8 we investigated the importance of the E-Step in our approach, by comparing to an ablation which randomly assigns tasks to policies at the start. These results showed that using random assignments performs worse, highlighting the importance of using related clusters of tasks. Here we will investigate how the difference between the return when using our EM method $G_{EM}$ or random assignments $G_{rand}$ changes depending on the number of tasks. When using a single policy or a policy for each task our method becomes identical to the baselines. We hypothesize that the difference should be maximal when using as many policies as there are true underlying clusters in the task set.

To test this hypothesis we perform experiments on our grid-world task set with 12 goals distributed to the four corners and show the return gap $G_{EM} - G_{rand}$ in Figure 12. The experiments confirm our hypothesis, showing that the return gap increases with the number of policies before reaching a maximum when it matches the true clusters at $n = 4$. Afterwards it starts to decrease.

