# OpenReview forum: "Unsupervised Task Clustering for Multi-Task Reinforcement Learning"
_ICLR.cc/2021/Conference — Reject_

### Official Review · AnonReviewer4 · 2020-10-24
**Interesting idea and simple approach; Additional discussion and investigation would be helpful.**

**Rating:** 6
**Confidence:** 5

**Review:**

### Summary:

The authors present a clustering-based approach to multi-task learning, specifically task-policy assignment, which iteratively learns a set of different policies for each of the subtasks. The aim is to solve issues with contradicting objectives of different tasks.

The clustering is achieved in an unsupervised way inspired by the EM algorithm, by iteratively evaluating the set of policies on all tasks, assigning tasks to policies based on their current task performance, and then training policies on their assigned tasks.

The proposed approach is evaluated on several continuous control and Atari multi-task RL problems, and compared to single policy and per-task policy baselines, as well as with a multi-head baseline inspired by an approach from the recent literature.
In order to show the importance of the clustering component of the approach, an ablation study is conducted.


### Review:

The paper proposes a simple and novel approach to task selection and policy learning in the multi-task learning problem setting. It provides proof of concept evaluations, as well as evaluations on some more complex environments where the task clustering is not obvious. I appreciate the discussion of the results and how the hypothesis relates to the observed outcomes.

Below, there are several suggestions that I believe would help improve the quality of the manuscript, and address some of the issues that should be clarified.

1) Regarding multi-task learning literature, it would be useful to mention the relationship with Quality-Diversity approaches:
 - Cully, Antoine, and Yiannis Demiris. "Quality and diversity optimization: A unifying modular framework." IEEE Transactions on Evolutionary Computation (2017)
 - Pugh, Justin K., Lisa B. Soros, and Kenneth O. Stanley. "Quality diversity: A new frontier for evolutionary computation." Frontiers in Robotics and AI (2016)

 As well as Deep RL approaches with skill/task conditioning:
 - Eysenbach, Benjamin, et al. "Diversity is all you need: Learning skills without a reward function." International Conference on Learning representations (2019).
 - Achiam, Joshua, et al. "Variational option discovery algorithms." arXiv preprint arXiv:1807.10299 (2018).

 It would greatly improve the strength of the manuscript to have comparisons with these state-of-the-art approaches as well, although I understand that this would require a significant effort from the authors.

2) Could you provide an example of these assumptions: “Note also that this formulation only relies on minimalistic assumptions. It is therefore applicable to a much broader range of settings than many sophisticated models with stronger assumptions.”

3) In the E step, the task $\tau_i$ is assigned to a policy $\pi_i$, which is used within the whole cluster $T_i$. There are N policies and K clusters, so it is not guaranteed that there is going to be one policy per cluster (i.e. $N=K$)? This part is not completely clear so it could be maybe explained.

 Related to this, how is the initialisation of the policies and clusters done, i.e. how are the number of initial policies and clusters selected? I guess the policies are manually set and this would be one of the hyperparameters that has to be tuned and is task-specific.
For the proof of concept tasks the number of policies can be set as the number of clusters that we can expect. Is this what is done in Bipedal tasks or is this an arbitrary number? For Atari experiments it is explicitly mentioned that n=4 and that the same is done for Bipedal tasks. This is a very important detail of the proposed approach and should be explained explicitly.

 Moreover, as I understand, at the beginning each task is attempted by each of the policies or they are assigned randomly. This explanation is missing.

4) It would be useful to provide some discussion about the stability of this approach, as in the beginning the policies could have tasks from different clusters assigned to them, but also the clusters change. I would assume that there is some burn-in period necessary.

5) The results on Atari for EM are not outperforming other baselines. Could you please comment on this? Could this be related to the number of policies set? It would be informative to evaluate the sensitivity of the EM algorithm to different numbers of initial policies.

 The conducted ablation study gives useful insights into the mechanisms of the approach. For the Corner-grid-world it makes sense that increasing n > 4 does not improve the performance, as there are 4 salient clusters (you could also mention this in the discussion). However, it is difficult to see a pattern in the effect of the number of policies for the Pendulum task. Therefore, investigating this in more detail, and in other environments as well, would greatly help understanding the approach better.

 One useful experiment would be to see how the performance gap between using randomly assigned clusters and learned clusters changes with respect to the initial number of policies.

6) Another useful metric would be evaluating the number of generated clusters and analysing how they relate to qualitatively different skills/tasks/behaviours.

7) Also, a comparison of computation time and required memory would be nice to show, with respect to SP and PPT for example, because additional policies need to be trained/stored. I assume, memory-wise and training-time-wise, EM is between SP and PPT.

OTHER COMMENTS:
- There are some links embedded in the pdf which are not visible and always link to the title page.
- What do you mean by “relationship of tasks of different clusters is **unconstrained**”, in which sense unconstrained ?
- How is the mean reward calculated for PPT in Figure 2? I assume it is done via evaluating each policy on the corresponding task, but this should be emphasised.
- Fig 5. right shows that SP converges faster than EM but then starts degrading, what could be an explanation for this?

---

> ### Author Response · Authors · 2020-11-13
> **Answer to Reviewer 4 (part 1)**
>
> Many thanks for the extensive and detailed review. We’ll address the concerns as they were enumerated:
>
> 1. The mentioned connection to quality-diversity (QD)-based approaches is indeed interesting. We did not consider it before. These works look into finding a diverse set of good solutions for a single objective, while our work focuses on finding solutions for each task in a set of a set of given tasks. One of the proposed benefits of QD is that it can overcome local optima, by being able to use the diverse solutions as “stepping stones” towards a global optimum. Similarly, [1] and [2] also first identify diverse skills and afterwards use the learned skills to solve the given task. While we do not implicitly encourage diversity in our approach, there is a clear connection in that our training leads to multiple good performing, distinct policies. This does sometimes lead to a policy trained on one task becoming the best on a task that it was not trained on. However, in our work this is more a side-effect than the proposed functionality.
> While our work is therefore not closely related to these approaches, there are certainly parallels and we will add a discussion of this topic to our related work section. As those approaches do not address the Multi-Task setting of our experiments, we do not compare against them in our experiments.
>
> 2. Example for assumptions in other approaches that limit their applicability:
> For example, there are some works that assume that the transition function is the same for all tasks, such as [3]. There are also works that look at cases where only the transition function is altered [4]. Further, earlier approaches (see e.g. [5]) require the underlying MDPs to be solvable in closed form. While such assumptions are of course reasonable in their settings, we address a more general setting in which we make no such assumptions.
>
> 3. Number of policies and number of clusters:
> We do not assume, in general, to have knowledge of the underlying number of clusters.
> We therefore determine the number of policies based on previous knowledge about the tasks, if available, or alternatively treat it as a hyperparameter.
> If the chosen number of policies is lower than the number of true clusters, the E-Step will assign multiple clusters to one policy, leading to a worse performance.
> This can for example be seen in Figure 7 on the left, where we use n=3 policies for the gridworld with K=4 true clusters.
> The number of policies in the Bipedal Walker and Pendulum experiments was chosen initially for consistency across environments and without tuning because we assumed that 4 would be enough.
> Regarding the initial assignment: We begin each trial of our approach with an E-Step; each task is attempted by each policy at the start.
> We will clarify both points in the text.
>
> 4. This is a good point, at the beginning of the training the performance of the policies varies a lot, leading to assignments changing frequently (see e.g. Fig.4 right). However, after a few trials, the policies improve and naturally specialize, leading to more stable assignments.
>
> 5. We assume that the reason that we are not outperforming baselines on Atari is because the Atari games are so diverse that our basic assumption of clusters, among which transfer is helpful, is violated. This is in line with results shown in related works, such as [6]. Our approach is still able to avoid a large part of the negative transfer, that completely hinders the baseline from learning a useful policy.
> We will clarify in the text that we do not expect to outperform PPT here.
> The ablation concerning the number of policies used in the pendulum tasks is indeed quite noisy, due to the stochastic nature of the pendulum tasks. We interpret the results as showing that a higher number of policies leads to a better final reward, as the policies can specialize to the specifics of each task. However, the results also show that a low number of policies suffices to reach a very good performance.
> The performance gap between using randomly assigned clusters and learned clusters can be seen to decrease with a higher number of used policies, as negative transfer is reduced with the smaller number of tasks assigned to each policy. We will add a new figure on this to the Appendix.
>
> 6. In the bipedal walker leg-length/obstacle spacing task set we can find that the clusters focus on the leg length and ignore obstacle spacing, while in the track and field tasks we usually find that one policy learns to walk/jump to the right, one learns to jump up and one learns to do the hurdles task. We show visualizations of the assignments in randomly picked trials in Figures 9 and 10 in the Appendix.

---

> > ### Author Response · Authors · 2020-11-13
> > **Answer to Reviewer 4 (part 2)**
> >
> > 7. Computation time and memory: As we perform the same number of policy updates per environment step in all approaches, the only real difference in complexity is caused by the E-step. As we find that E-Steps are mainly necessary during the earlier part of training, they could be omitted or reduced later on. Overall, even in the current approach the added computation requirement is small and not significant compared to the computation required for training the policies.
> > The total required memory is dominated by the replay buffers used to store previous transitions. As we use a replay buffer per policy, the required memory of EM is somewhere between SP and PPT. The required memory of PPT is the same as that of the Multi-Head baseline, as both use a separate buffer per task.
> >
> > OTHER COMMENTS:
> >
> > Thanks for the comment, we will remove the links.
> >
> > Q: What do you mean by “relationship of tasks of different clusters is unconstrained”, in which sense unconstrained?”
> > A: While we assume that tasks in one cluster benefit from transfer, tasks in different clusters may or may not benefit from transfer.
> >
> > We indeed evaluate each policy on its corresponding task in PPT, we will clarify this in the text.
> >
> > Fig 5 SP quick convergence and subsequent degradation:
> > We hypothesise that this is due to the fact that in an initial stage, the tasks can be easily distinguished with different measurements (LIDAR sensors) corresponding to different tasks. This leads to a fast convergence initially. However, as training progresses tasks require more specialization and start competing over the capacity of the network. This can lead to negative interference which can even lead to an overall decrease in performance. However, this is just a hypothesis.
> >
> >
> > [1] Eysenbach, Benjamin, et al. "Diversity is all you need: Learning skills without a reward function." International Conference on Learning representations (2019).
> >
> > [2] Achiam, Joshua, et al. "Variational option discovery algorithms." arXiv preprint arXiv:1807.10299 (2018).
> >
> > [3]  A. Barreto et al., “Successor features for transfer in reinforcement learning,” NeurIPS (2017).
> >
> > [4] K. Lee, et al. “Context-aware Dynamics Model for Generalization in Model-Based Reinforcement Learning,” ICML (2020).
> >
> > [5] A. Wilson et al. “Multi-task reinforcement learning: a hierarchical Bayesian approach” ICML (2007)
> >
> > [6] M. Hessel et al. “Multi-task Deep Reinforcement Learning with PopArt.” AAAI 2019

---

### Official Review · AnonReviewer3 · 2020-10-27
**Technical contributions seem to be limited.**

**Rating:** 5
**Confidence:** 3

**Review:**

In this paper, the authors present a new multi-task reinforcement learning (RL) algorithm. Since In general, the relationships between tasks is unknown a-priori, directly applying classical multi-task learning approaches that assume all tasks are related, could suffer from negative transfer. The authors propose to cluster tasks into disjoint groups: The proposed algorithm iterates through steps of assigning tasks to specific policies and training each policy only based on the respective assigned tasks (clusters). In the experiments, the authors compare their algorithm with two single-task learning baselines (SP: a single policy for all tasks and PPT: a policy per task) and a recent multi-task RL algorithm of Eramo et al. 2020 on Pendulum, Bipedal Walker, and Atari problems.

While multi-task learning was extensively studied in supervised learning, its application to RL has only recently begun to gain attention. This paper contributes by a new multi-task RL algorithm that addresses the negative transfer issue arising when one tries to apply multi-task learning strategies across partially unrelated tasks. However, I am not sure if this paper is ready for ICLR in the current form, for the reasons given below

- Limited novelty: Negative transfer in multi-task learning has been previously studied, mainly in supervised learning but recently, also in RL. Specifically, task clustering approaches have been previously examined although as far as I am aware, they were used only in supervised learning.
Even though the proposed EM strategy is not a trivial application of existing approaches, the technical novelty seems to be limited. Enhancing the practical relevance of the proposed algorithm via an extensive empirical study (involving multiple state-of-the-art approaches and challenging real-world problems) could help. However --->
- Limited experiments: Two simple baselines are not multi-task learning algorithms, and Eramo et al.’s approach was not designed to address the negative transfer problem. I think the experiments can be improved by including comparisons with more, state-of-the-art algorithms including these discussed in the last two paragraphs of Section 2. Yu et al. 2020 Sharma et al. 2018 Hesse et al. are designed to address the negative transfer issue, thus they should provide better baselines than SP, PPT, and Eramo.
- ATARI experiments (Figure 6) seem to suggest that when the individual tasks are not related, (hence multi-task learning is not beneficial), applying the proposed algorithm can degrade the performance from the PPT, still suffering from the negative transfer. Their results are much better than Eramo et al. 2020, but the latter method was not designed for negative transfer cases.

Section 4 describes well the operations performed by the proposed EM algorithm. Still, explicit equations on the E and M steps would help reading.

Thank the authors for their responses. I read through the responses from the authors and comments from the other reviewers. I would maintain my initial rating: I think this paper will benefit significantly from a major revision, either strengthening the theoretical contributions or improving empirical validations (by actually performing extensive comparisons with existing algorithms that are designed to handle negative transfer problems).

---

> ### Author Response · Authors · 2020-11-13
> **Response to Reviewer 3**
>
> Many thanks for the detailed review. In the following we will answer the comments one by one.
>
>
> Novelty:
> We agree that the topic of negative transfer in multi-task learning has been studied before, as has the concept of task clustering. However, most previous work on multi-task learning in reinforcement learning (RL) is quite specific to an application, has strong assumptions about the structure of the tasks or is only applicable to small problems where an underlying Markov decision Process (MDP) can be fully solved. Instead of seeing the simplicity of our approach as limited novelty, we see it as an advantage as it can very easily be combined with any underlying RL method and many proposed Multi-Task RL approaches. Nonetheless, we will strengthen our discussion of the related work on this topic, also in accordance with the comments of the other reviewers
>
> Baselines:
> We agree that comparisons to other approaches specifically meant to combat negative transfer would strengthen the paper While all of the mentioned approaches can be combined with our approach and are therefore somewhat orthogonal, we see that a direct comparison to Gradient Surgery [1] would be beneficial, as their main aim is similar to ours. We have therefore implemented this method and performed experiments on the pendulum task set. We observe a very high variance in the results. Some trials perform similar to our method, though converging a bit more slowly, while a few trials (3/12) perform similar to a random policy. We are not entirely sure why, but one reason could be the relatively small network in our experiments (2 hidden layers) compared to the one in their trials (6 hidden layers). This might lead to the task-specific gradients being more important, while also being affected by gradient surgery. We will add these results in the upcoming revision.
>
>
> The other approaches [2,3] address different problems that arise in multi-task RL. While Sharma et al. [2] look at tasks with varying difficulty, Hessel et al. [3] look at tasks with varying reward scales. Both approaches can be easily combined with our approach, we therefore do not provide a direct comparison.
>
> Atari Experiments:
> It is true that our method performs worse than PPT if the tasks are not related. This was expected, as it violates our assumption of it being possible to decompose the task set into clusters of similar tasks. To completely avoid negative transfer, the number of policies should not be less than the number of underlying true clusters. However, if the tasks are very diverse, each task forms its own cluster and as many policies as tasks would be necessary, at which point our approach becomes almost identical to PPT.
> We will improve the discussion of this section in the text.
>
> Clarity in Section 4:
> We will explicitly add the equations for the two steps to make the writing more clear.
>
> [1] T. Yu et al. “Gradient Surgery for Multi-Task Learning,” NeurIPS (2020).
>
> [2] S. Sharma et al. “Learning to Multi-Task by Active Sampling,” ICLR (2018).
>
> [3] M. Hessel et al. “Multi-task Deep Reinforcement Learning with PopArt.” AAAI (2019).

---

### Official Review · AnonReviewer1 · 2020-10-31
**Review #1**

**Rating:** 5
**Confidence:** 5

**Review:**

This paper proposes a multi-task RL algorithm that leverages unsupervised task clustering. The authors propose to initialize a number of policies, cluster each task based on its performance on different policies, and train each policy with data coming from tasks within the cluster. The paper shows that such kind of an EM style clustering can lead to better performance than single-task training and be more sample efficient more training each task independently on both tabular settings, continuous control experiments, and Atari.

For pros, I think the method is relatively simple yet effective based on the empirical evaluations, which is a plus. Moreover, the empirical results seem to suggest that clustering can indeed recover the natural clustering or help performance when natural clustering is not obvious to humans, which could make the method valuable. The paper is also well-written and easy to understand.

However, I do have a few concerns about this paper, which I will list as follows:

1. It seems that there are several old MTRL papers [1, 2] that also consider task clustering using EM-style approaches, but are not discussed and compared in this paper.

2. The paper seems more like combining clustering methods that are used in multi-task supervised learning and RL and such approaches are also explored in prior works as mentioned in 1, which makes the contribution a bit derivative. Moreover, the idea of learning separate policies seems a bit contradictory to the goal of the MTRL, which is learning a single policy that can tackle all tasks. The solution proposed in the paper, which is to learn multiple policies, kind of defeats the purpose of MTRL. Also, since the algorithm for complex tasks is TD3, is the value function also learned per-cluster? It would much more appealing if the authors can somehow distill the multiple policies into a single policy that can tackle all tasks, as in Distral [3].

3. For the experiments, do you use the same network for SP and PPT as in EM? In that case, doesn't EM have more parameters than SP?  I think it would be fairer if the total number of parameters is the same for all three methods. Moreover, it would be important to show the performance on benchmarks with tasks that are more distinct such as Meta-World, which could justify the strength of the clustering approach in settings where natural clustering is near impossible.

4. Finally, given the relatively less novel method, it would be interesting to see if there's any theoretical insight into this method, but it is currently missing.

Overall, given the pros and cons listed above, I would vote for a weak reject of this paper.

[1] Lazaric, Alessandro, and Mohammad Ghavamzadeh. "Bayesian multi-task reinforcement learning." 2010.
[2] Li, Hui, Xuejun Liao, and Lawrence Carin. "Multi-task Reinforcement Learning in Partially Observable Stochastic Environments." Journal of Machine Learning Research 10.5 (2009).
[3] Teh, Yee, et al. "Distral: Robust multitask reinforcement learning." Advances in Neural Information Processing Systems. 2017.
[4] Yu, Tianhe, et al. "Meta-world: A benchmark and evaluation for multi-task and meta reinforcement learning." Conference on Robot Learning. 2020.

---

> ### Author Response · Authors · 2020-11-13
> **Response to Reviewer 1**
>
> Many thanks for their detailed comments. We will address the comments as they were enumerated.
>
> 1. The two mentioned papers are indeed related and we will add a discussion of  them to our related work section. We thank you for the pointers. However, both approaches use EM to learn/infer the policy, not to cluster tasks. Task clustering is done via a Dirichlet Process. [1] uses a hierarchical Bayesian approach to infer the parameters of a linear value function while [2] infers the parameters of the decision state Markov decision processes (MDPs) corresponding to the partially observable MDPs investigated. It is unclear how either of those approaches would scale to complex tasks with possibly infinite state spaces. Regarding [1]: a linear value function might not be enough to capture complex dynamics or requires non-trivial feature engineering. Regarding [2]: in their approach every update scales quadratically with the size of the decision state space and (in the best case) linear in the number of collected data points. This becomes prohibitively expensive in complex environments.
>
>
> 2. The value functions are also learned per cluster, yes.
> Regarding your point that the learning of separate policy seems contradictory to the goal of Multi-Task RL: We see this as mainly an issue of terminology. We consider one agent as consisting of multiple policies and thereby deviate from the common phrasing where policy and agent are used interchangeably. Our goal in this work is to quickly reach a good performance on all given tasks, regardless of how many underlying networks are used. We do not see an issue in practice with having multiple policies as an end-result.
> While we do require knowledge of which task the agent is currently performing, this is also the case in related work that uses a single policy with multiple outputs, such as in  [4,5]. Note that Distral [3], while appealing as a concept, does not resolve the negative transfer issue.
>
>
>
> 3. This is a good point. While we did control the parameter count for the other baselines, we did not do so for SP and PPT in the experiments. We found this comparison to be  unfair towards PPT, as it is thereby limited to a very small network, which leads to a poor performance even if just having to learn a single task. We did, however, rerun our SP baseline with the network size increased to the same number of parameters as in our EM approach. This did not change the results, as the low performance is caused by the contradictory reward functions, not by a lack of representational capacity. We will update the figures and text in our submission to show these results.
> While Meta-World is a very interesting and challenging benchmark, we opted to instead perform experiments in the ALE, due to the proprietary nature of the physics engine underlying Meta-World. Nevertheless, the ALE tasks are significantly more diverse than the Meta-World tasks. We also presented the variations on the Bipedal-Walker task as an alternative, in which the robot is largely the same but the environment and reward differ.
>
>
> 4. While we did look into a theoretical justification of our approach, we felt that it did not provide significant new insights beyond the discussion of our approach already in the paper. We therefore omitted it from the submission but will consider adding to the Appendix for the final version.
>
> [1] Lazaric, Alessandro, and Mohammad Ghavamzadeh. "Bayesian multi-task reinforcement learning." 2010.
>
> [2] H. Li et al. "Multi-task Reinforcement Learning in Partially Observable Stochastic Environments." Journal of Machine Learning Research 10.5 (2009).
>
> [3] Y. Teh, et al. "Distral: Robust multitask reinforcement learning." NeurIPS (2017).
>
> [4] S. Sharma et al. “Learning to Multi-Task by Active Sampling,” ICLR (2018).
>
> [5] M. Hessel et al. “Multi-task Deep Reinforcement Learning with PopArt.” AAAI (2019).

---

### Official Review · AnonReviewer2 · 2020-10-31
**Informal algorithm closely related to previously published approaches**

**Rating:** 5
**Confidence:** 4

**Review:**

The authors propose to approach multi-task RL problems in which tasks may differ considerably in transition functions/dynamics and reward functions as well as in the action space through task clustering. Specifically, tasks are modeled as belonging to separate task clusters defined by the return obtainable by individual policies i. All policies are evaluated on a single task and the relative cumulative discounted rewards over some iterations determines the assignment of tasks to policies. Simulations show the advantage in terms of number of training iterations on several tasks compared to a selection to other related algorithms.

MTRL is an interesting and relevant field for ICLR in which research has been quite active for years.

The authors’ algorithm is loosely inspired by EM in that the unknown task assignments of individual learners are thought of as a latent task assignment/ clustering variable. As the assignments to clusters are not soft/probabilistic, this seems almost more inspired by k-means than by EM.

The algorithm is a straightforward extension of previous approaches. Clustering of tasks is also an idea, that has been pursued in the field, just one example is Wilson, Fern, Ray, Tadepalli (2007). There also seems to be a close connection to the mixture of experts (Jacobs, Jordan, 1991) in which the responsibility values are used to select individual learners. This framework has also been used in control settings. The authors may want to consider relating their algorithm to this literature.

The authors do not comment on the selection of the number n of policies, that needs to be selected before learning, similarly to the number of clusters in mixture models for clustering. It would be desirable to at least comment on how to select this parameter.

The authors may wish to comment on the complexity of their algorithm. For the case in which a moderate number cluster with highly related tasks in each cluster the proposed algorithm may work well, if the number n of policies is selected accordingly.

The simulations show empirically, that for tasks in which the similarity between tasks in a cluster is high, i.e. the changes in policy required between tasks in the cluster are small, learning is faster. This is to be expected by design of the considered learning problems. The authors also provide learning results in which the advantage is not given, as in the ALE tasks.

Overall, this is a quite informal presentation of an algorithm that is closely related to several previously published algorithms. The empirical evaluations are promising for specific multitask settings.

---

> ### Author Response · Authors · 2020-11-13
> **Response to Reviewer 2**
>
> Many thanks for the extensive review. Please find our answer in the following:
>
> K-Means:
> Regarding your note that in its current shape our approach seems more inspired by k-means than by EM, due to the assignments not being soft. This is a good point. During our experiments we also evaluated such soft-assignments, however, they did not seem to significantly improve performance.  We thus omitted these evaluations in our submission.
>
> Related-Work:
> We thank you for the pointers. Be assured that they will be included in the first revision (coming soon). As a preview:
> Wilson et al. develop a hierarchical Bayesian approach in which they use a dirichlet process to cluster the distributions from which they sample Markov decision processes (MDP) that might align with the task at hand. They then solve the sampled MDP and use the resulting policy to gather data from the environment and refine the posterior distributions for a next iteration. Our approach is different in that it does not assume explicit underlying MDPs. Our neural network based approach has thus the potential to scale to MDPs with large or infinite state spaces which cannot be solved in closed form.
> The Mixture of experts (Jacobs, Jordan, Nolan and Hinton 1991) framework goes in another direction. Note that this work and the work that describes a EM training procedure for it (see “Hierarchical mixtures of experts and the EM algorithm”, Jordan and Jacobs 1993) is proposed for the supervised setting and considers a gating which is input dependent. Our approach is different in two distinct aspects. (1) we focus on reinforcement learning where clustering has to be done over MDPs, which is more complex than clustering independent samples as in supervised learning. (2) Our final method performs a “hard” clustering. This has the advantage that once we know the task assignment all but the best performing policy/expert can be discarded. While responsibility values, as proposed by Jacobs and Jordan, help with mitigating interference, they do not prevent it. Our approach is more explicit in this regard in that, once the underlying clusters are found, interference is completely eliminated.
> Regarding the use of a mixture of experts for control: A quick literature research revealed several papers [1-5] and we thank for the pointer. However, these approaches either require per time step labels [1-4] or an optimal target trajectory [5]. All of them are trained with supervised learning. Furthermore, while [1,2] look at a control problem with multiple modes, the main aim in [3-5] is to address discontinuities in the control problem. A paper which deploys an architecture similar to a mixture of experts and investigates multi-task reinforcement learning is [6], which we already included in our initial write up of our paper.
> These other references are important and we will be adding them to our discussion of the related work. However, to us they seem quite distinct to our method and contribution as discussed above.
>
> Selection of number of policies:
> If we have some previous knowledge about the given tasks we propose to use this to select the number of policies.
> If this is not clear, we propose to treat the number of policies as a hyper-parameter, similarly to how it would be done in clustering approaches. We did not explicitly tune this hyper-parameter in our experiments, as we found that using n=4 policies worked well in all our experiments. We also found that using slightly more policies than necessary is not problematic, as shown in our ablation experiment.
>
> Complexity:
> The complexity of the evaluation-step scales with number of tasks $|\mathcal{T}|$ and number of policies $n$, i.e.,  $n\cdot|\mathcal{T}|$ evaluations are necessary to assign policies. Therefore, the reviewer is correct that the evaluation step will be expensive if a large number of tasks or policies is used. However, we found that assignments are quite stable after the first few iterations. Therefore, a hierarchical clustering with cluster refinement might reduce this evaluation overhead in future work that considers a larger number of tasks.
>
> [1] A competitive modular connectionist architecture (Jacobs and Jordan 1991)
>
> [2] Learning Piecewise Control Strategies in a Modular Neural Network Architecture (Jacobs and Jordan 1993)
>
> [3] Learning Fine Motion by Markov Mixtures of Experts (Meila and Jordan, 1995)
>
> [4] Mixtures of Controllers for Jump Linear and Non-linear Plants (Cacciatore and Nowlan, 1994)
>
> [5] Discontinuity-Sensitive Optimal Control Learning by Mixture of Experts (Tang and Hauser, 2019)
>
> [6] Attentive Multi-Task Deep Reinforcement Learning (Bram et al., 2019)

---

### Author Response · Authors · 2020-11-17
**Updated Version**

We have uploaded a new version of our submission in accordance with the suggestions of the reviewers.
 * We extended the related work with discussions of referred papers.
 * We added a comparison to the gradient-surgery method [1].
 * We adjusted the size of the SP baseline and updated the results.
 * We clarified the description of our approach based on the points raised by the reviewers.
 * We added an additional ablation to the appendix showing how the performance gap between our approach and random assignments to clusters varies with the number of used policies $n$.

[1] T. Yu et al. “Gradient Surgery for Multi-Task Learning,” NeurIPS (2020).

---

### Decision · Program_Chairs · 2021-01-07
**Final Decision**

**Decision:**

Reject

**Comment:**

This paper considers multi-task RL from the perspective of an unsupervised clustering of different tasks with an EM-like algorithm. The idea is evaluated on several simple and ATARI domains.
We thank the reviewers for their detailed responses and revision. This work still seems a little preliminary in its current form. While the empirical results seem promising, it is generally felt that it would benefit from more extensive experiments, including further comparisons to other approaches and exploring the effects of the hyperparameters on tasks with much larger numbers of clusters. It would also be beneficial to provide some theoretical results, particularly with respect to negative transfer.